# Phylogeny of *Rhynchium* and Its Related Genera (Hymenoptera: Eumeninae) Based on Universal Single-Copy Orthologs and Ultraconserved Elements

**DOI:** 10.3390/insects14090775

**Published:** 2023-09-20

**Authors:** Min Dai, Shu-Lin He, Bin Chen, Ting-Jing Li

**Affiliations:** Chongqing Key Laboratory of Vector Insects, Institute of Entomology and Molecular Biology, College of Life Science, Chongqing Normal University, Chongqing 401331, China; 2021110513042@stu.cqnu.edu.cn (M.D.); shulinhe@cqnu.edu.cn (S.-L.H.); bin.chen@cqnu.edu.cn (B.C.)

**Keywords:** Eumeninae, low-coverage whole genomes, universal single-copy orthologs [USCO], ultraconserved elements [UCE], phylogeny

## Abstract

**Simple Summary:**

The eumenine wasps of the genus *Rhynchium* Spinola, 1806; *Allorhynchium* van der Vecht, 1963; *Anterhynchium* de Saussure, 1863; and *Pararrhynchium* de Saussure, 1855 are related and sometimes hard to tell apart from each other. In this study, we first reconstructed the phylogenetic relationships of these genera based on universal single-copy orthologs and ultraconserved elements extracted from 10 newly sequenced low-coverage whole genomes. The results showed that *Allorhynchium* and *Lissodynerus* are distinct from the other four taxa. The genus *Rhynchium* was recovered as monophyletic, whereas *Anterhynchium* was recovered as paraphyletic, with *Anterhynchium* (*Dirhynchium*) as a sister to *Rhynchium* and hence deserving a separate genus *Dirhynchium*; and within the genus *Pararrhynchium*, *P. septemfasciatus feanus* and *P. venkataramani* were separated, not clustered on a branch. It is suggested that the genus *Lissodynerus* should be restituted as a valid genus, not a synonym of the *Pararrhynchium.* The results are consistent with previous eumenine mitochondrial genome phylogenetic analyses. This paper confirms the feasibility of low-coverage whole genome eumenine wasp phylogenetics and provides a reference for subsequent research in Eumeninae.

**Abstract:**

The subfamily Eumeninae is a large group of fierce predatory insects that prey mainly on the larvae of Lepidoptera pests. Because of the highly similar morphologies of the genus *Rhynchium* and its related genera in the subfamily, including *Rhynchium* Spinola, *Allorhynchium* van der Vecht, *Anterhynchium* de Saussure, *Pararrhynchium* de Saussure, it is essential to delineate their relationships. A previous phylogenetic analysis based on mitochondrial genomes suggested the inconsistent relationships of these genera under traditional classification based on morphological characters. In this study, we first used single-copy orthologs [USCO] and ultraconserved elements [UCE] extracted from 10 newly sequenced low-coverage whole genomes to resolve the phylogenetic relationships of the above genera. The newly sequenced genomes are 152.99 Mb to 211.49 Mb in size with high completeness (BUSCO complete: 91.5–95.6%) and G + C content (36.31–38.76%). Based on extracted 5811 USCOs and 2312 UCEs, the phylogenetic relationships of *Rhynchium* and its related genera were: ((*Allorhynchium* + *Lissodynerus*) + (*Pararrhynchium* + (*Anterhynchium* + (*Dirhynchium* + *Rhynchium*)))), which was consistent with the mitochondrial genome results. The results supported the genus *Rhynchium* as monophyletic, whereas *Anterhynchium* was recovered as paraphyletic, with *Anterhynchium* (*Dirhynchium*) as a sister to *Rhynchium* and hence deserving generic status; In addition, in the genus *Pararrhynchium*, *P. septemfasciatus feanus* and *P. venkataramani* were separated, not clustered on a branch, just as *P. septemfasciatus feanus* was not together with *P. striatum* based on mitochondrial genomes. Since *Lissodynerus septemfasciatus*, the type species of the genus *Lissodynerus*, was transferred to *Pararrhynchium*, it is considered that the genus *Lissodynerus* should be restituted as a valid genus, not a synonym of *Pararrhynchium*.

## 1. Introduction

The subfamily Eumeninae (Hymenoptera: Vespidae) is a large group of fierce predatory insects that prey mainly on the larvae of Lepidoptera pests and can be used for biological control [1,2]. It contains seven genera with roots *rhynchium* worldwide, including *Allorhynchium* van der Vecht, 1963, *Anterhynchium* de Saussure, 1863, *Emeryrhynchium* Gusenleitner, 2007, *Gibberhynchium* Gusenleitner, 2002, *Pararrhynchium* de Saussure, 1855, *Rhynchium* Spinola, 1806, and *Xenorhynchium* van der Vecht, 1963. They are restricted to the Old World [3,4], with four, *Rhynchium*, *Anterhynchium*, *Pararrhynchium* and *Allorhynchium*, found in China. Hereafter, we refer to these four genera as the *Rhynchium* genera group in this study. The *Rhynchium* genera group has similar morphology, with a body size ranging from 8.9 mm to 15.1 mm, relatively stout, and slightly narrower first than the second gastral segment [5,6,7,8,9,10,11,12,13,14]. Because of the highly similar morphologies of these genera, it is sometimes challenging to tell them apart from each other, and therefore delineating their relationships is a significant exploration.

The phylogenetic relationships within the *Rhynchium* genera group have been discussed in a previous study based on morphological features and mitochondrial genomes [15], but uncertainties and conflicting classification schemes remain among genera and subgenera and on the taxonomic status of certain species. Firstly, the taxonomic status of *Anterhynchium* (*Dirhynchium*) is controversial. *Dirhynchium* van der Vecht, 1963 has long been regarded as a subgenus of *Anterhynchium* [8,9,10,11] based on morphology. This, however, conflicts with the mitochondrial phylogenetic analysis which suggested that the subgenus *Dirhynchium* should be elevated to genus [16]. Meanwhile, the taxonomy of wasps in *Anterhynchium* is still poor. As Selis and Carpenter systematically revised African *Anterhynchium* and did a phylogenetic analysis based on morphological characters to move several species from this genus to other genera [17], are there more revisions including the division of subgenera in this genus that needs to be improved in other regions? Secondly, the taxonomic status of the species *Pararrhynchium septemfasciatus feanus* (Giordani Soika 1941) remains uncertain. *Pararrhynchium septemfasciatus feanus* (Giordani Soika 1941) was transferred from the genus *Lissodynerus* with the synonymy of the genus *Lissodynerus* as *Pararrhynchium* by Carpenter and Brown based on morphological characters [18]. Originally, *P*. *septemfasciatus feanus* (Giordani Soika 1941) was described as *Ancistrocerus septemfasciatus var. feanus* Giordani Soika, 1941 in the genus *Ancistrocerus*, and then was combined within the genus *Lissodynerus* by Giordani Soika [19,20], which was supported by Tan et al. and Li et al. [5,21]. In a systematic taxonomic collection of the subfamily Eumeninae in China, Tan et al. placed *P. septemfasciatus feanus* in the genus *Lissodynerus* based on the prestigma of the forewing, which is longer than half the length of the pterostigma [5]. Meanwhile, in our previous study of the subfamily Eumeninae based on mitochondrial genomes [16], the species *P. septemfasciatus feanus* ran out of the genus *Pararrhynchium*. Resolving these above uncertainties requires more in-depth research. 

With advancing sequencing technologies and increasing genomic data, low-coverage whole genome sequencing is becoming a popular technique in phylogenetic studies due to its simple sample preparation and low cost [22]. Universal single-copy orthologs (USCOs) are selected from OrthoDB orthologous groups that contain genes present as single-copy orthologs in at least 90% of the species and are of particular importance in providing the basis to infer species phylogenetics [23]. Due to their slow evolutionary rate, less saturation than other markers, and not crossing over with most types of paralogous genes, Ultraconserved elements (UCEs) are another set of increasingly used markers for phylogenetic analysis [24,25]. Therefore, reconstructing phylogenetic relationships with USCOs and/or UCEs has been successfully applied to a range of species, such as springtails, bees, rodents, and monocots [26,27,28,29]. At present, although systematic phylogenetic analysis of all major lineages of Hymenoptera based on protein-coding genes has been conducted [30], phylogenetic studies of the subfamily Eumeninae based on nuclear data are still poorly. In this paper, we aim to investigate the phylogenetic relationships of *Rhynchium* and its related genera in the subfamily Eumeninae using low-coverage whole genome sequencing with USCOs and UCEs, which could also serve as new molecular markers for the subfamily Eumeninae.

## 2. Materials and Methods

### 2.1. Sampling and Sequencing

Here, we sampled 10 species of the *Rhynchium* genera group with two species for each of the genera *Rhynchium*, *Allorhynchium*, and *Pararrhynchium*, and four species of the two subgenera *Anterhynchium* and *Dirhynchium* in the genus *Anterhynchium* (detail information is shown in Table 1, more information see Appendix A). Specimens of the 10 species were netted, smothered in collection tubes, immersed in 95% ethanol, and stored at −20 °C at the Institute of Entomology and Molecular Biology, Chongqing Normal University. The head and gaster were removed to reduce contamination of the intestinal flora; and the remaining tissue was sent to Novogene Bioinformatics Technology Co., Ltd. (Tianjin, China) for sequencing. Genomic DNA was extracted and purified using the DNeasy DNA Extraction Kit (Qiagen). The qualified libraries were pooled and sequenced on Illumina platforms with PE150. 

We chose two species in Vespinae as outgroups, including the genomic assemblies of *Dolichovespula media* (GCA_911387685.1) and *Vespa crabro* (GCA_910589235.2) from NCBI.

### 2.2. Genome Assembly and Matrix Generation

BBTools v38.96 [31] was used for quality control. The duplications were removed by clumpify.sh; low-quality reads were trimmed with bbduk.sh; reads shorter than 15 bp and containing more than 5 consecutive Ns after trimming were discarded; remaining reads were then normalized with bbnorm.sh to reduce the complexity and speed up the assembling. The reads were assembled de novo and scaffolded by using SPAdes v3.15.5 [32]; the gaps were filled by GapCloser v1.12 [33] in the SOAPdenovo2 to replace base N with base A\T\G\C. The completeness of the assemblies was evaluated by using BUSCO v5.4.3 with the Hymenoptera database (https://busco-data.ezlab.org/v5/data/lineages/hymenoptera_odb10.2020-08-05.tar.gz, accessed on 28 June 2021).

Two different marker sets (USCO and UCE) were extracted separately from the 10 genomes. Universal single-copy orthologs (USCOs) were extracted with BUSCO v5.4.3 [34] against Hymenoptera reference gene sets (*n* = 5991). USCO amino acid sequences were used for subsequent analyses. Each protein-coding gene sequence was aligned with the L-INS-I method in MAFFT v7.490 [35], and each alignment was trimmed with BMGE [36] to remove unreliable homologous regions and decrease compositional heterogeneity. Those alignments that violated the stationary, reversible, and homogeneous (SRH) assumptions [37] were further excluded using ‘--symtest’ implemented in IQ-TREE v2.0 [38]. The trimmed alignments were subsequently concatenated using FASConCAT-g v1.05 [39]; three USCO amino acid matrices (USCO50, USCO90, and USCO100) with the completeness of 50%, 90%, and 100% were constructed, representing the lowest ratio of taxa for all partitions (see Appendix A). Ultraconserved element probes have been designed in several insect orders, including Hymenoptera. The Hymenoptera 2.5 kv2 bait set [40], containing 31,827 baits targeting 2590 loci, was used as the reference probe set. The UCEs of the 10 assembled genomes were extracted by PHYLUCE v1.6.3 [41]. Three UCE matrices, including UCE60, UCE90, and UCE100, were generated in a similar way to the construction of USCO matrices, including MAFFT alignment, BMGE trimming, FASConCAT-g concatenation, and IQ-TREE filtering for SRH model violations, for subsequent analyses (see Appendix A).

### 2.3. Phylogenetic Analyses

We used partitioned maximum likelihood (ML), heterologous model, and multi-species coalescent model (MSC) methods to reconstruct the phylogenetic relationships of the *Rhynchium* genera group. For the partitioned ML analyses, we used the MODELFINDER [42] to select the most appropriate substitution model for each partition by employing the relaxed hierarchical clustering algorithm ‘-rclusterf 10’ in IQ-TREE [43]. The best protein substitution model for the USCO matrices was ‘LG + I + G’, and the best substitution model for the UCE matrices was ‘GTR + I + G’. The full IQ-TREE command line used was ‘iqtree2 -s example.phy -p example.nex -m MFP + MERGE --symtest-remove-bad -B 1000 -alrt 1000 --prefix -rcluster 10’. To avoid the influence of heterotachous evolutionary sequences on phylogenetic inference, we also used the General Heterogeneous Evolution on a Single Topology (GHOST) [44] model in IQ-TREE with ‘-m LG + FO*H4’ for USCO amino acids and with ‘-m GTR + FO*H4’ for UCE nucleotides. Simultaneously, 1000 replicates of SH-aLRT [45] and 1000 replicates of UFBoot2 [46] were performed for all ML analyses to obtain species trees. For the multi-species coalescent model analyses, individual gene trees were estimated from the previous steps of matrix generation, and ASTRAL-III v5.7.1 [47] was used with default parameters to infer species trees, with the advantage of highly accurate support values and fast computation speed [48]. Although support values such as bootstrap or posterior probability give statistical confidence in the species tree topology, there may still be significant underlying gene tree discordance [49] even for branches with 100% support. Therefore, genealogical consistency between species and gene trees was quantified with the gene concordance factor (gCF) and the site concordance (sCF) using IQ-TREE.

In addition, site heterogeneity has been highlighted as more important than protein-wise heterotactic (partitioning) in phylogenomic analyses [50]. To avoid long-branch attraction (LBA) artifacts, we also applied site-heterogeneous models in ML inference. The posterior mean site frequency (PMSF) [51] model was performed for each USCO matrix by specifying a profile mixture model LG + C20 + F + G in IQ-TREE.

### 2.4. Hypothesis Testing

Following the phylogenetic analyses, one node showed inconsistent results with analyses based on information from individual genes (ASTRAL and partitioned ML) and analyses of the different matrices using profile mixture models (GHOST and PMSF). Gene tree heterogeneity, such as incomplete lineage sorting, is a known source of erroneous inference in summary coalescent methods [52]. We suspected that the discordance between our results was due to uninformative genes in gene-wise partitioning and gene trees. First, using likelihood-mapping (LMAP) [53] as implemented in IQ-TREE, we evaluated whether individual genes contained sufficient information to distinguish between different topologies at the two competing nodes. For the recalcitrant node, four clusters of taxa were defined (Vespinae, *Rhynchium* s.s., *Pararhynchium venkataramani*, and *Allorhynchium* s.s.) (Figure 1). The topologies 1 and 2 were of interest; the topology 3, which was equally conceivable given the unrooted tree with four clusters but had not been in any of the earlier analyses, could have been recovered. Then, we tested the resultant two alternative topologies (T1, T2) with the matrix UCE90 using approximately the unbiased (AU), weighted Kishino–Hasegawa (WKH), and weighted Shimodaira–Hasegawa (WSH) tests under the partitioned model ‘-m GTR + I + G’ in IQ-TREE: T1, (*Allorhynchium s.s.* + *Pararrhynchium venkataramani*) + remaining taxa; T2, *Allorhynchium s.s.* + (*Pararrhynchium venkataramani* + remaining taxa). The tree estimated from previous ASTRAL inference on matrix UCE90 was used as the guide tree and the starting tree.

### 2.5. Divergence Time Analyses

Divergence times were estimated using MCMCTree in PAML v4.9j [54] based on the nucleotide sequences of UCEs. We performed divergence analyses for three matrices (UCE60, UCE90, and UCE100) representing three different data sizes (2,396,740, 2,159,911, and 1,933,308 sites, respectively, Table 2) to evaluate the potential influence of the data size on the branch lengths. For input trees, topology T2 estimated from previous ASTRAL analyses was preferred, and calibration fossils were searched on paleobiodb (https://www.paleobiodb.org/, accessed on 5 August 2023). The fossil record of *Rhynchium andrei* (61.6–59.2 Ma) [55] was used as a calibration point for the crown age of *Rhynchium*. To reduce the computational burden, approximate likelihood calculation and ML estimation of branch lengths was performed by using ‘usedata = 2 and 3’ [56]. We used the independent rates clock model (clock = 2) and the GTR substitution model (model = 7) to calculate the Hessian matrices. We set the parameters of the mean substitution rate and the rate drift as ‘rgene_gamma = 2 20 1’ and ‘sigma2_gamma = 1 10 1’. The estimation was run for 1 billion MCMC generations sampled every 1,000,000 generations, with the first 0.2 million as burn-in. Convergence was checked using Tracer v1.7.2 [57] to ensure the Effective Sample Size was over 200.

## 3. Results

### 3.1. Genome Assembly and Matrix Generation

We sequenced approximately 15 G raw data for each sample, with 72 to 100 sequence coverage. The sizes of the assemblies range from 152.99 Mb (*Pararrhynchium septemfasciatus feanus*) to 211.49 Mb (*Allorhynchium chinense*). The number of scaffolds ranges from 15,712 to 187,058, the N50 of the assemblies from 23.39 to 670.26 kb, the max read length from 842.5 kb to 4723.7 kb, and the GC content from 36.31% to 38.76%. Additional statistics, including average read coverage, the number of scaffolds, assembly size, maximum read length, and N50, are provided in Table 1.

The completeness of genome assemblies was 91.50–95.60%, of which 0.10–0.20% were duplications, and 1.40–3.60% were fragmented (Figure 2 and Table 1). The sequences of the same ortholog were merged into a FASTA file, resulting in 5811 USCOs with amino acid sequences ranging from 2,423,609 to 3,551,689. For UCE, a total of 2312 UCEs with 2,585,262 nucleotide sites were extracted. After sequence alignment, screening, and trimming, the six matrices used for phylogenetic analysis included 4398–5560 USCOs and 1726–2312 UCEs with 2,423,609–3,327,612 amino acid sites and 2,159,911–2,475,217 nucleotide sites, respectively (Table 2).

### 3.2. Phylogenetic Relationships among the Rhynchium Genera Group

Two topologies of the *Rhynchium* genera group were inferred based on the matrices and inference models, with the majority of nodes consistent except for the position of *Pararrhynchium venkataramani* in both topologies (see Appendix A). Under the first topology (T1), *Pararrhynchium venkataramani* was sister to a clade comprising *Pararrhynchium septemfasciatus feanus,* and *Allorhynchium*; under the second topology (T2), *Pararrhynchium venkataramani* was sister to a clade comprising (*Anterhynchium* + (*Dirhynchium* + *Rhynchium*)). The partitioning analyses of all matrices and the GHOST model and PMSF model of all USCO matrices supported topology 1 (T1). However, the GHOST model of UCE matrices supported topology 2 (T2), and ASTRAL analyses of UCEs and USCOs (USCO100 excluded) recovered topology 2 (T2) (Table 3). All tree files are available in the Appendix A.

Furthermore, we conducted an LMAP test for the relationships of *Pararrhynchium venkataramani* with others. The T2 was supported by USCOs matrices (more than 79.5%) and UCEs matrices (100%) (Figure 3). Meanwhile, T2 was strongly supported by all hypothesis tests (AU, WKH, and WSH) and T1 was strongly rejected (Table 4). Combining LMAP tests and hypothesis tests, the results supported the monophyletic *Anterhynchium* as a sister group to a clade comprising two genera *Dirhynchium* and *Rhynchium*. Meanwhile, *Pararrhynchium septemfasciatus feanus* and *Pararrhynchium venkataramani* were separated, not clustered on a branch (Figure 4).

### 3.3. Molecular Dating of the Rhynchium Genera Group

All the runs of data matrices have ESS values greater than 200, and a strong correlation (R2 > 0.99) of the convergence plots indicates good convergence between parallel runs (Figure 5). The MCMCTree results (Figure 6) show that the most recent common ancestor (MRCA) of the *Rhynchium* genera group was inferred to be the Cretaceous (95.74 million years (Ma), 83.09–107.58 at 95% highest posterior density (HPD)). The crown groups of *Allorhynchium* and *Pararrhynchium septemfasciatus feanus* originated during the Cretaceous–Paleogene (71.93 Ma, 41.27–96.98); the crown age of *Allorhynchium* was estimated at the Paleogene–Quaternary (13.98 Ma, 1.42–33.35); the origins of *Pararrhynchium* and the clade containing *Dirhynchium*, *Rhynchium*, and *Anterrhynchium* occurred during the Cretaceous (88.35 Ma, 76.29–100.04); the origins of the clade comprising *Dirhynchium* and *Rhynchium* also occurred during the Cretaceous–Paleogene (71.47 Ma, 62.96–80.86); the crown age of *Rhynchium* falls at the Paleogene (60.24 Ma, 59.15–61.54), consistent with the timing of the fossil node; *Dirhynchium* and *Anterrhynchium* occurred during the Paleogene–Quaternary (14.57 Ma, 1.11–36.5; 13.98 Ma, 1.42–33.35, respectively).

## 4. Discussion

### 4.1. Sources of Gene Tree Discordance

In our studies, phylogenetic results were incongruent depending on the inferred models. Partitioned ML, GHOST (USCO matrices), and PMSF analyses favored topology 1, while GHOST (UCE matrices) and a coalescent approach based on gene trees recovered topology 2. In the two topological trees, the support values such as bootstrap were high, but the gene concordance factor (gCF) and the novel site concordance factor (sCF) values (gCF < 30, sCF < 35, see Appendix A) for the inconsistent node was relatively low, demonstrating that systematic errors due to gene tree conflicts are the main reason for the inconsistency of the two phylogenetic trees.

Gene tree conflicts in closely related species are commonly due to genealogical discordances, such as ILS [58,59,60], which have been detected in hominids, pines, cichlids, finches, and grasshoppers [61,62,63,64,65]. MSC implemented by ASTRAL incorporates a higher degree of biological realism than the conventional concatenation-based models, which is expected to produce more plausible reconstructions [47]. In the present study, most of the ASTRAL analyses (except the USCO100 matrix) support T2, *Pararrhynchium venkataramani* was sister to a clade comprising (*Anterhynchium* + (*Dirhynchium* + *Rhynchium*)). In addition, accounting for the heterogeneity of substitution patterns between sites and genes in proteins [51], we also applied site-heterogeneous models to mitigate possible long-branch attraction (LBA) artifacts. Interestingly, the phylogenetic trees obtained from the site-heterogeneous model (partitioned and GHOST) and the site-homogeneous model (LG + PMSF(C20)) based on the USCO matrices were congruent, suggesting that there’s little LBA bias in this study. The inconsistent phylogenetic trees obtained by partitioned ML, GHOST, MSC, and PMSF models, were tested by LMAP tests, which support T2 by both the USCO and UCE matrices, so ILS is the main source of conflict in the gene tree of this study.

UCEs have grown in appeal among hymenopteran systematists who employ hybrid enrichment or occasionally transcriptome sequencing techniques [66,67,68], while the application of USCOs as tools for phylogenetic inference in Hymenoptera is still in its infancy. The successful application of USCOs and UCEs in bees [27] suggests that low-coverage whole genomes could resolve uncertainty in phylogenetic analyses of Hymenoptera, providing a new set of molecular markers. We used both USCOs and UCE markers from low-coverage whole genomes for phylogenetic reconstruction and finally obtained consistent results with maximum support for all nodes. Once again, it has been proven that phylogenetic reconstructions in Hymenoptera using USCOs and UCEs from low-coverage whole genomes are feasible.

### 4.2. Phylogenetic Relationships

The best-supported topology shows the clade consisting of *Pararrhynchium*, *Anterhynchium*, *Dirhynchium*, and *Rhynchium* is monophyletic; *Allorhynchium* and *Lissodynerus* are more distantly related to the other four genera.

In one clade, a monophyletic *Anterhynchium* is the sister to a clade comprising *Dirhynchium* and *Rhynchium*. *Anterhynchium*, *Dirhynchium*, and *Rhynchium* form a monophyletic lineage that shares a common feature: propodeum and metanotum at different levels in lateral view [5,8,9,10,11,14,15]. This supports the elevation of *Dirhynchium* to a valid genus, as it is proposed based on a mitogenomic analysis [16] and corroborated by morphological differences. Current diagnostic characters of *Dirhynchium* are as follows: basal narrow part of the first gastral sternum with dense transverse striae through the whole width, the third and following gastral terga very coarsely punctate at the base, male mandible not deeply emarginated on the inner side. However, the diagnostic characters of the nominotypical subgenus *Anterhynchium* are distinctly different from *Dirhynchium*: narrow basal part of the first gastral sternum smooth or with only a narrow median band of fine, short, and shiny transverse striae, gastral terga only finely punctate at the base, male mandible deeply emarginated on the inner side near the middle [8,9,10,11]. Therefore, we propose a new genus, *Dirhynchium* van der Vecht, 1963.

In the phylogeny, the species *Pararrhynchium septemfasciatus feanus* is more closely related to *Allorhynchium* van der Vecht, 1963 rather than *Pararrhynchium venkataramani*, indicating the species does not belong to the genus *Pararrhynchium*. Considering *P. septemfasciatus* was transferred from *Lissodynerus* as a synonym *Pararrhynchium* [18], and *Lissodynerus septemfasciatus* was the type species of the genus *Lissodynerus*, the genus *Lissodynerus* should be restituted as a valid genus, not a synonym of the genus *Pararrhynchium*. The result is consistent with the analysis of mitochondrial genomes [16]. Further analyses with more specimens are needed to investigate whether the rest of the species previously included in the genus *Lissodynerus* should be all returned to *Lissodynerus* or some of them should be transferred to the genus *Pararrhynchium* as it already happened in the study of Carpenter & Brown [18].

### 4.3. Effects of Continental Drifts on the Distribution of Rhynchium Genera Group

The *Rhynchium* genera group is distributed in the Old World only, including the Australian, Ethiopian, Oriental, and Palearctic regions. The most recent common ancestor (MRCA) of the *Rhynchium* genera group was inferred in the Cretaceous (95.74 Ma), which is consistent with the results of the evolutionary history of the Hymenoptera [30]. The *Rhynchium* genera group might have spread in various plates according to the migration history of the Old World. During the Cretaceous, the Pangaea [69] split up and Africa collided with Europe, leading to the formation of the Alps [70]. In addition, following the separation of the Gondwanaland during the Late Jurassic (~135 Ma) to India and Oceania, the Eurasian subcontinent was formed after the collision between India and Asia in Southern Tibet in the Eocene (56–33.9 Ma) [71]. Consequently, the continents of the Old World except Australia had been connected from the Cretaceous–Paleogene (84–45 Ma) until the present day [71], which might be the reason that the *Rhynchium* genera group is distributed in the Old World. Moreover, the presence of many islands between Australia and the Eurasian subcontinent, such as the Philippine Islands and New Guinea, might raise the dispersal of the *Rhynchium* genera group to Australia through these islands just as the *Ropalidia* species groups in the vespid wasps [72,73].

Meanwhile, the ocean might be a limiting factor in the distributions of the *Rhynchium* genera group in the New World. During the Late Jurassic/Early Cretaceous (~110 Ma), the second breakup of Gondwana rifting of South America and Southern Africa [74] led to the formation of the South Atlantic basin [75,76]. In addition, during the Jurassic (~169.7 Ma) the Pacific Ocean basin was born and is by far the oldest preserved oceanic crust within the global oceans [77]. The estimated age of the MRCA of *Rhynchium* genera group in the Cretaceous is later than the formation of the ocean, which might consequently lead to the no distribution of this group in the New World. 

## 5. Conclusions

Overall, our phylogenetic analyses based on different combinations of data matrices and models yielded consistent trees. However, the inconsistency of one node between analyses reaffirms the necessity of a large data set and model selection for phylogenetics [26]. Combining methodological suitability with morphological evidence, the topology recovered using the MSC model was preferred. *Rhynchium*, *Dirhynchium*, *Anterhynchium*, and *Pararrhynchium* comprise a monophyletic group; *Allorhynchium* and *Lissodynerus* are distinct from the other four taxa. Of these, it was corroborated that *Dirhynchium* van der Vecht, 1963 should be upgraded to a valid genus, and *Lissodynerus* Giordani Soika, 1993 should be restituted as a valid genus, not a synonymy of *Pararrhynchium* de Saussure, 1855. Considering the limitations of the sample, the attribution of the other species previously placed in *Lissodynerus* requires further investigation. This study is the first phylogenetic study using USCOs and UCEs extracted from low-coverage whole genomes in Eumeninae, providing a reference for further studies on the phylogeny of the subfamily Eumeninae. 

## Figures and Tables

**Figure 1 insects-14-00775-f001:**
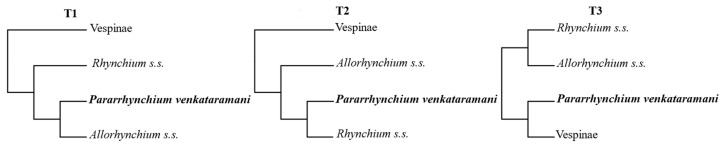
Likelihood mapping analyses (LMAP). Possible topologies for the position of *Pararrhynchium venkataramani* based on four clusters of taxa.

**Figure 2 insects-14-00775-f002:**
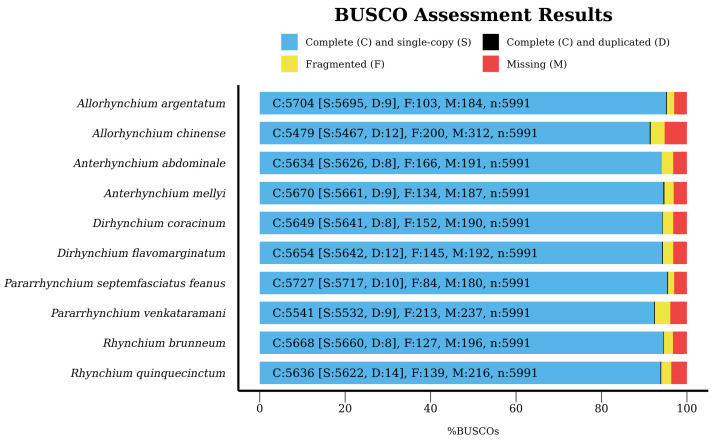
BUSCO completeness: complete (C), complete single-copy (S, light blue), complete duplicated (D, black), fragmented (F, yellow), and missing (M, red).

**Figure 3 insects-14-00775-f003:**
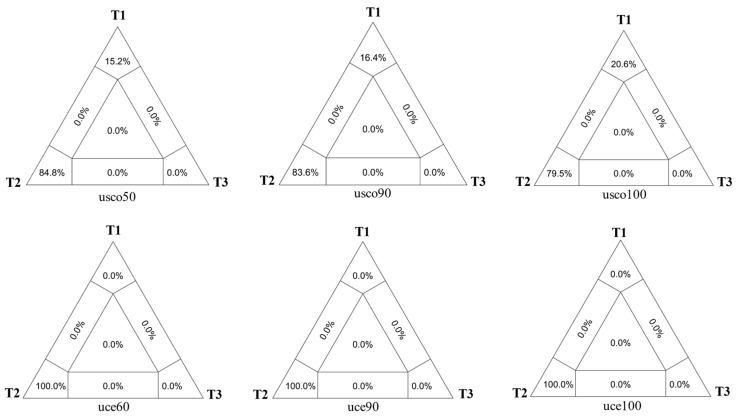
Likelihood maps for the position of *Pararrhynchium venkataramani* from two matrices with contrasting distributions of quartets.

**Figure 4 insects-14-00775-f004:**
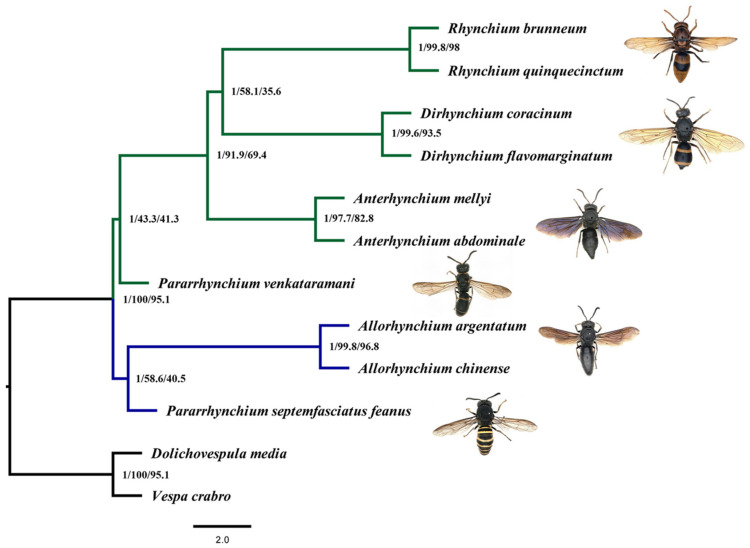
Phylogeny of the *Rhynchium* genera group inferred from matrix UCE90 using the MSC model implemented in ASTRAL. Node labels show BPP/gCF/sCF.

**Figure 5 insects-14-00775-f005:**
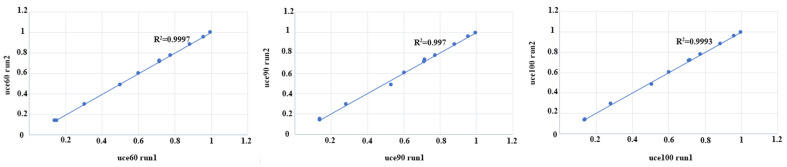
The convergence plot of MCMCtree result.

**Figure 6 insects-14-00775-f006:**
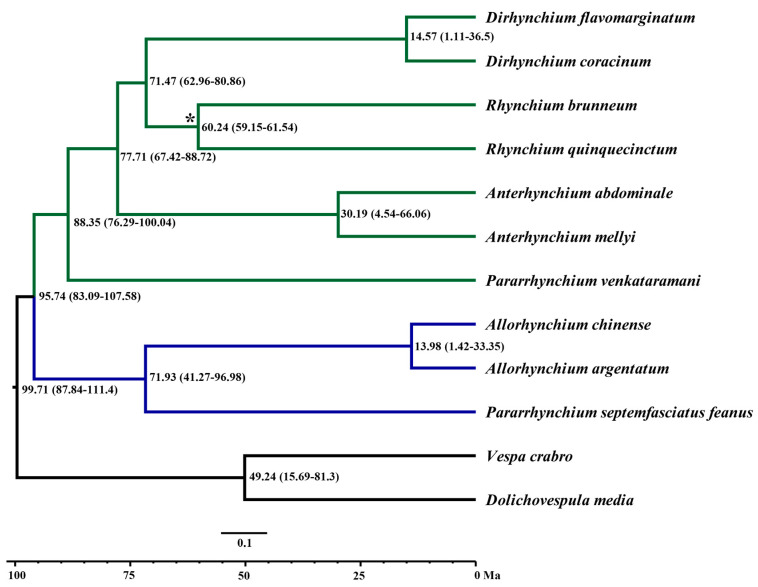
The *Rhynchium* genera group divergence time estimation performed by using MCMCTree on the ASTRAL topology. Node numbers and node bars represent 95% CIs of the estimated divergence times integrated from all MCMC runs. Asterisk mark node constrained with the fossil.

**Table 1 insects-14-00775-t001:** Summary for 10 newly-sequenced low-coverage whole genomic samples.

Species	Accession Number	Average Read Coverage (X)	BUSCO Completeness (%)	The Number of Scaffolds	Assembly Size (Mb)	Max Read Length (kb)	N50 Scaffold (kb)	GC (%)
*Rhynchium brunneum*	SAMN36845277	78.85	94.6	42,348	194.79	1170.58	134.32	37.24
*Rhynchium quinquecinctum*	SAMN36845333	78.26	94.0	48,619	196.27	1281.6	105.73	37.24
*Allorhynchium argentatum*	SAMN36845335	84.95	95.3	49,682	180.81	894.85	160.54	37.27
*Allorhynchium chinense*	SAMN36845336	72.63	91.5	187,058	211.49	318.76	23.39	36.31
*Anterhynchium* (*A*.) *abdominale*	SAMN36845337	83.57	94.0	60,405	183.79	586.96	63.68	38.61
*Anterhynchium* (*A*.) *mellyi*	SAMN36845339	90.69	94.7	23,487	169.37	842.5	112.22	38.66
*Anterhynchium* (*Dirhynchium*) *flavomarginatum*	SAMN36845338	86.10	94.4	23,366	178.40	786.82	74.28	37.83
*Anterhynchium* (*Dirhynchium*) *coracinum*	SAMN36845334	86.89	94.3	21,837	176.77	739.09	68.99	37.69
*Pararrhynchium septemfasciatus feanus*	SAMN36845341	100.40	95.6	15,712	152.99	4723.7	670.26	38.53
*Pararrhynchium venkataramani*	SAMN36845340	85.82	92.5	51,762	178.97	728.23	51.56	38.76

**Table 2 insects-14-00775-t002:** Summary of USCOs (amino acid) and UCEs (nucleotide) matrices.

Matrix	Average Missing Taxa per Locus (%)	Number of Loci	Number of Sites	Missing Sites (%)	Average Locus Length
USCO50	2.99	5714	3,467,928	3.19	606.92
USCO90	1.05	5115	2,945,110	1.18	575.78
USCO100	0	4398	2,423,609	0.01	551.07
UCE60	2.56	2138	2,396,740	2.56	1121.02
UCE90	0.77	1928	2,159,911	0.85	1120.29
UCE100	0	1726	1,933,308	0.24	1120.11

**Table 3 insects-14-00775-t003:** Summary of topological hypotheses recovered under different analyses.

Matrix	Partitioning	GHOST	ASTRAL	PMSF
USCO50	T1	T1	T2	T1
USCO90	T1	T1	T2	T1
USCO100	T1	T1	T1	T1
UCE60	T1	T2	T2	-
UCE90	T1	T2	T2	-
UCE100	T1	T2	T2	-

**Table 4 insects-14-00775-t004:** Results of the hypothesis tests.

Topologies	log*L*	delta*L*	bp-RELL	*P*-WKH	*P*-WSH	*P*-AU
T1	−8,408,349.19	964.1	0 *	0 *	0 *	2.88 × 10^−6^
**T2**	**−8,407,385.093**	**0 ***	**1**	**1**	**1**	**1**

Best-fit hypotheses are given in boldface. 0 * means that the value < 0. AU, approximately unbiased; RELL, resampled estimated log-likelihood; WKH, weighted Kishino–Hasegawa; WSH, weighted Shimodaira–Hasegawa.

## Data Availability

The data presented in this study are openly available from the National Center for Biotechnology Information at https://www.ncbi.nlm.nih.gov (accessed on 5 August 2023), accession numbers: PRJNA1002480 (https://dataview.ncbi.nlm.nih.gov/objects?linked_to_id=PRJNA1002480, accessed on 5 August 2023).

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
