# Peer review of "Phylogeny of Rhynchium and Its Related Genera (Hymenoptera: Eumeninae) Based on Universal Single-Copy Orthologs and Ultraconserved Elements"

_insects, 2023, doi:10.3390/insects14090775_

Round 1

Reviewer 1 Report

This is an interesting manuscript with phylogenetic reconstruction of several closely related and taxonomically complicated genera of wasps. In my opinion, its general scientific quality is high. I, however, cannot evaluate molecular phylogenetic methods since I am not a specialist, but taxonomic application of the results can be improved. I have two main suggestions.

1. Authors should restitute Lissodynerus as valid genus instead of expressing doubts about this. Lissodynerus septemfasciatus is the type species of the genus Lissodynerus. If it is not related to Pararrhynchium (as was clearly shown in the manuscript), it means that not only this species should be moved to another genus but the genus Lissodynerus should be restituted as a valid genus, not a synonym of Pararrhynchium. May be it is worthy to check if all species previously placed in Lissodynerus are indeed Lissodynerus or may be some are Pararrhynchium (or even from other genera), but anyway the synonymy of Lissodynerus under Pararrhynchium should be rejected.

2. I strongly suggest to enlarge the introduction by adding a reference to a recent revision of African Anterhynchium by Selis and Carpenter in Zootaxa, https://doi.org/10.11646/zootaxa.5233.1.1 with a discussion on the taxonomic changes made by those authors. Particularly, they did a phylogenetic analysis based on morphological characters. They also moved several species from this genus to other genera that means that the taxonomy of this group of wasps is still poorly elaborated.

Please see the attached file for further details. There are also some minor suggestions there.

Reviewer 2 Report

Dear authors,

I have diligently perused your manuscript, finding it both captivating and of significant value to fellow researchers. Your comprehensive coverage of all aspects was commendable, and I found no instances of ambiguity. The clarity of your presentation was exceptional. Congratulations on your remarkable contribution, and I encourage you to continue your endeavours in exploring various facets of this taxonomic group.

Best regards,

The language is generally clear and concise, which is appropriate for professional communication.

Author Response

Thank you for your encouragement, we will continue more in-depth research of this taxonomic group. Based on the opinions of three reviewers and editor, we have added the corresponding literature. 

Reviewer 3 Report

The manuscript is well-written and the methods and results sound. Please correct a number of minor grammatical errors indicated in the attached.

Well-written with few grammatical errors.
